# Newton-coupled Dual-Teacher Semi-supervised Learning Framework

Hongyang He [* 1]   Xinyuan Song [* 2]   Yan Zhong [3]   Daizong Liu [4]   Xuanyu Liu [1]   Victor Sanchez [1]

## Abstract

Most semi-supervised learning frameworks rely on a single teacher that transfers zero-order supervision through pseudo-labels, constraining the student to imitate categorical outputs without perceiving the loss geometry. This design often leads to unstable optimization and limited generalization under scarce labels. To address this shortcoming, we propose **TTN** (Two-Teachers Newton-guided Learning), a dual-teacher framework that integrates complementary supervision from MAE and DINOv3 and optimizes the student through a Newton step update. The two teachers provide multi-scale structural and semantic cues whose pseudo-labels and local Hessians are fused by confidence weighting, forming a unified second-order supervision signal. The student updates parameters preconditioned by the fused curvature, enabling stable convergence and geometry-consistent learning. TTN consistently improves over existing single-teacher and consistency-based semi-supervised learning methods on ImageNet, CIFAR-10, SVHN, and STL-10, demonstrating that combining multi-view self-supervised teachers with curvature-guided optimization yields robust and efficient semi-supervised learning.

## 1. Introduction

Recent semi-supervised learning (SSL) approaches such as Pseudo-Label (Lee et al., 2013), Mean Teacher (Tarvainen & Valpola, 2017), UDA (Xie et al., 2020), FixMatch (Sohn et al., 2020), and FlexMatch (Zhang et al., 2021) have achieved remarkable progress by coupling pseudo-label generation with consistency regularization. Despite their suc-

---
*Equal contribution [1]Department of Computer Science, University of Warwick, Coventry, United Kingdom [2]Emory University, Atlanta, GA, United States [3]Peking University, Beijing, China [4]Wuhan University, Wuhan, China. Correspondence to: Daizong Liu <daizongliu@whu.edu.cn>.

*Proceedings of the $43^{rd}$ International Conference on Machine Learning*, Seoul, South Korea. PMLR 306, 2026. Copyright 2026 by the author(s).

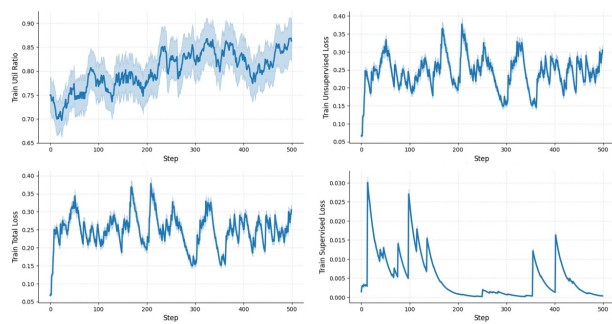

*Figure 1.* Training dynamics of FixMatch on CIFAR-100 with 400 labeled samples.

cess, these frameworks share two fundamental limitations in their knowledge transfer mechanism. First, the information propagated from teacher to student is essentially *zero-order*: the teacher supervises the student only through categorical pseudo-labels or confidence-weighted probabilities, conveying what the prediction should be but not how it should behave in the local geometry of the loss surface. Consequently, the student model merely imitates the teacher's output without perceiving curvature, smoothness, or uncertainty in the surrounding optimization landscape. As shown in Fig. 1, the pseudo-label utilization ratio in FixMatch remains close to one throughout training, implying that most unlabeled samples are treated as confidently labeled under deterministic supervision, while the unsupervised loss exhibits strong oscillations, revealing that the model is repeatedly perturbed by non-flat decision regions and must implicitly smooth its loss to maintain consistency. These observations highlight the inherent zero-order nature of current SSL supervision, where the teacher provides outcome-level guidance but no geometric awareness.

The second limitation is the structural design of most SSL frameworks; namely, nearly all existing methods adopt a *single-teacher, single-student* architecture, which further limits the richness of the supervision signal (Lee et al., 2013; Tarvainen & Valpola, 2017; Sohn et al., 2020; Zhang et al., 2021; Blum & Mitchell, 1998; Qiao et al., 2018; Grandvalet & Bengio, 2004; Laine & Aila, 2016; Chen et al., 2023; Cai et al., 2022; He et al., 2025c; Xie et al., 2025; He & Hong, 2025; Hong et al., 2026a; He et al., 2026d; 2025b; 2026c;a;b; Hong et al., 2026b). This design choice makes the optimization trajectory overly dependent on one teacher

representational bias, restricting the diversity and robustness of pseudo-label generation. Whether the teacher is built from a self-ensembling model, an EMA-updated network, or a pre-trained backbone, the information passed to the student originates from a single perspective of the feature space. Such single-source supervision amplifies the zero-order constraint—it enforces imitation of one model's categorical predictions without exposing the student to multi-view or multi-scale geometric cues that could promote better generalization. In practice, we observe that this single-source configuration also leads to unstable optimization. When only one frozen encoder is used as the teacher, the training process often exhibits gradient explosion or collapse, accompanied by sharp oscillations in the unsupervised loss and curvature magnitude (Fig. 4). This instability results in inferior accuracy and inconsistent convergence, as confirmed by our ablation results (Table 4). These phenomena suggest that a single-teacher model fails to provide sufficiently rich or geometrically consistent guidance, reinforcing the need for multi-view supervision (Blum & Mitchell, 1998; Qiao et al., 2018) and second-order regularization.

To overcome these limitations, we advocate a paradigm shift in the way supervision is transmitted in SSL. Rather than constraining the student through fixed, zero-order pseudo-labels produced by a single teacher, we propose that the teacher(s) should communicate richer information about the optimization geometry and provide complementary structural perspectives. Specifically, our framework introduces two self-supervised teachers—MAE (He et al., 2022) and DINOv3 (Oquab et al., 2023)—that guide the student from distinct yet synergistic representations: MAE captures fine-grained, patch-level regularity, while DINOv3 models global semantic smoothness. By allowing the student to learn jointly from these complementary views, the supervision evolves from discrete output imitation to a geometry-aware, multi-scale transfer of knowledge.

## 2. Related Works

**Semi-Supervised Learning.** SSL has evolved from classical probabilistic and geometric formulations that exploit unlabeled data via low-density separation, graph regularization, or manifold assumptions (Blum & Mitchell, 1998), to modern deep learning paradigms. Early approaches provide strong theoretical foundations but suffer from limited scalability and noise sensitivity. With deep neural networks, self-training and pseudo-labeling have emerged as dominant strategies, where models iteratively label unlabeled data (Lee et al., 2013). Although confidence-based filtering and curriculum scheduling improve label quality (Sohn et al., 2020; Zhang et al., 2021; Berthelot et al., 2019), single-view supervision often leads to unstable gradients and biases.

Modern SSL frameworks extend pseudo-labeling through

consistency regularization and teacher–student mechanisms (Tarvainen & Valpola, 2017; Xie et al., 2020; Sohn et al., 2020), enforcing prediction agreement across perturbations. Mean Teacher (Tarvainen & Valpola, 2017) and FixMatch (Sohn et al., 2020) perform well under limited labels, but deterministic thresholding can amplify noisy pseudo-labels. Recent methods such as FlexMatch (Zhang et al., 2021), UDA (Xie et al., 2020), and Meta Pseudo Label (Pham et al., 2021) introduce adaptive thresholds and uncertainty-aware supervision, yet the learning signal remains categorical and largely detached from optimization geometry. Consequently, most SSL methods optimize only probabilistic outputs, without accounting for feature-space structure or local curvature. Transformer-based SSL frameworks, including Semi-ViT (Cai et al., 2022) and Semi-ViM (He et al., 2025c), further improve scalability but still rely primarily on zero-order pseudo-label signals.

**Multi-View and Dual-Teacher Learning.** Co-training frameworks exploit multiple data views or networks to enhance pseudo-label reliability through mutual agreement (Blum & Mitchell, 1998). Deep co-training extensions (Qiao et al., 2018) adopt multiple encoders, adversarial perturbations, or meta-learned filtering to stabilize collaboration between models. However, existing multi-teacher approaches remain zero-order—they aggregate categorical predictions but ignore the underlying geometry of each teacher's representation. Combining heterogeneous self-supervised teachers, such as reconstruction-based MAE (He et al., 2022) and contrastive encoders like DINO and DINOv2/v3 (Caron et al., 2021; Oquab et al., 2023; He et al., 2020), offers richer supervision but demands an effective mechanism for aligning their feature scales and uncertainty distributions. More recent unified SSL paradigms such as TRiCo (He et al., 2025a) have begun to incorporate game-theoretic or meta-regularized objectives, but they still lack explicit modeling of curvature or second-order information in the teacher–student interaction.

## 3. Proposed Framework: TTN

TTN introduces multi-scale supervision and curvature-aware optimization into the teacher–student paradigm. TTN consists of three interacting components: two complementary teachers, $\{T_A, T_B\}$, pre-trained under self-supervised objectives (MAE and DINOv3), and a curvature-guided student, $S$, that learns from both labeled and unlabeled data (see Fig. 2). The two teachers provide distinct yet complementary signals: MAE emphasizes fine-grained patch-level structures, while DINOv3 captures global semantic consistency. At each training step, both teachers generate soft pseudo-labels for the same unlabeled batch. Their predictions are combined through a fusion module that emphasizes agreement in the predictive distributions and downweights

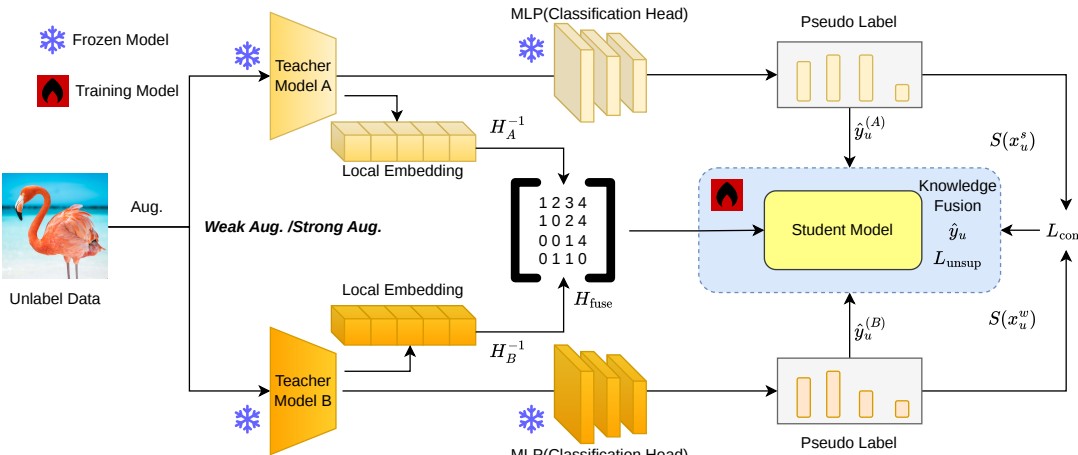

*Figure 2.* **Overview of the proposed TTN framework.** Two self-supervised teachers provide complementary global and local supervision on unlabeled data. Their pseudo-labels and local embedding curvature are fused through an uncertainty-aware module to guide the student. The student performs Newton-Step style updates.

conflicting predictions. The fused pseudo-labels form the supervisory target for the student, while the teachers' local embedding curvature further modulates the student's optimization dynamics.

Instead of updating the student with conventional gradient descent, TTN employs a curvature-aware rule that approximates Newton-style updates. This mechanism adapts the student's learning direction to the geometric cues provided by both teachers, encouraging smoother and more stable convergence. The training loop operates iteratively: the teachers extract pseudo-labels and curvature cues from weakly augmented samples; the fusion module aggregates them into a unified supervision signal; and the student performs a curvature-modulated update that balances global and local knowledge. Through this interaction, TTN transforms the classical zero-order teacher–student learning into a second-order, geometry-aware process, allowing the student to learn not only from the teachers' predictions but also from their structural priors.

TTN departs from prior single-teacher pipelines, where supervision is restricted to categorical pseudo-labels detached from the optimization geometry. By contrast, TTN establishes an interactive dual-teacher structure that continuously informs the student's update trajectory with complementary geometric knowledge.

### 3.1. Dual-Teacher Knowledge Fusion

TTN integrates complementary supervision from two self-supervised teachers, $T_A$ (MAE) and $T_B$ (DINOv3), which process the weakly augmented unlabeled sample $x_u^w$ to produce pseudo-labels and local curvature estimates:

$$\hat{y}_u^{(A)} = T_A(x_u^w), \qquad \hat{y}_u^{(B)} = T_B(x_u^w), \qquad (1)$$

together with their embedding Hessians, $H_A$ and $H_B$. The two teachers provide distinct yet complementary information: $T_A$ captures fine-grained structural cues, while $T_B$ emphasizes global semantic regularities.

To achieve interpretable and stable supervision, TTN fuses the two pseudo-labels according to their predictive confidence. The confidence scores $\omega_A$ and $\omega_B$ are respectively defined for the two teachers as the maximum predicted probability of the softmax output:

$$\omega_A = \max_c \hat{y}_u^{(A)}[c], \qquad \omega_B = \max_c \hat{y}_u^{(B)}[c], \qquad (2)$$

where $[c]$ denotes the $c$-th component of the softmax probability vector, i.e., the predicted probability assigned to class $c \in \{1, \dots, C\}$. These scores reflect the reliability of the teachers on the current sample and are normalized as $\tilde{\omega}_A = \frac{\omega_A}{\omega_A + \omega_B}$, $\tilde{\omega}_B = 1 - \tilde{\omega}_A$. Since the softmax distribution belongs to an exponential family, the Kullback–Leibler (KL)-barycenter between two distributions with weights $\tilde{\omega}_A$ and $\tilde{\omega}_B$ corresponds to a convex combination in the log-probability domain:

$$\log \hat{y}_u = \tilde{\omega}_A \log \hat{y}_u^{(A)} + \tilde{\omega}_B \log \hat{y}_u^{(B)}. \qquad (3)$$

This yields the normalized weighted geometric mean:

$$\hat{y}_u = \frac{(\hat{y}_u^{(A)})^{\tilde{\omega}_A} (\hat{y}_u^{(B)})^{\tilde{\omega}_B}}{\sum_c (\hat{y}_u^{(A)}[c])^{\tilde{\omega}_A} (\hat{y}_u^{(B)}[c])^{\tilde{\omega}_B}}. \qquad (4)$$

The fusion adaptively emphasizes high-confidence, mutually consistent predictions while down-weighting uncertain or conflicting ones, resulting in a more interpretable and reliability-aware supervisory target.

Curvature information is combined following the same confidence weighting principle. Because the student update

employs the inverse Hessian, $H^{-1}$, as a second-order preconditioning metric that approximates the local curvature of the loss landscape, we fuse the inverse Hessians from the two teachers using a weighted harmonic mean on the manifold of symmetric positive-definite matrices:

$$H_{\text{fuse}} = \left( \tilde{\omega}_A H_A^{-1} + \tilde{\omega}_B H_B^{-1} \right)^{-1}. \qquad (5)$$

This weighting mitigates the effect of ill-conditioned curvature estimates from less confident teachers and yields a coherent second-order geometry that reflects both local precision from MAE and global smoothness from DINOv3. The resulting $\hat{y}_u$ and $H_{\text{fuse}}$ jointly provide confidence-aware, geometry-consistent supervision for the student's Newton-guided optimization.

### 3.2. Newton-step Student Optimization

After obtaining the fused pseudo-label, $\hat{y}_u$, and curvature matrix, $H_{\text{fuse}}$, the student $S$ updates its parameters through a Newton-step style optimization process. This stage aligns the student's learning trajectory with the geometric priors distilled from both teachers. Instead of conventional gradient descent, TTN introduces a second-order correction that adapts the update direction according to the fused curvature, ensuring stable and geometry-consistent learning.

The student minimizes a total loss consisting of a supervised term on labeled data and an unsupervised term on fused pseudo-labels:

$$\mathcal{L}_{\text{total}} = \mathcal{L}_{\text{sup}} + \mu\, \mathcal{L}_{\text{unsup}}, \qquad (6)$$

where $\mu$ balances the contribution of the unlabeled component. The supervised loss is computed using cross-entropy on labeled samples, while the unsupervised loss enforces consistency between the student's prediction on strongly augmented inputs, $x_u^s$, and the fused pseudo-label $\hat{y}_u$ generated by the dual teachers.

During optimization, the gradient of $\mathcal{L}_{\text{total}}$ with respect to the student's parameters, $\theta_s$, is preconditioned by the inverse fused curvature. The parameter update follows a second-order rule approximating a **Newton step**:

$$\theta_s \leftarrow \theta_s - \eta\, H_{\text{fuse}}^{-1}\, \nabla_{\theta_s} \mathcal{L}_{\text{total}}, \qquad (7)$$

where $\eta$ is the learning rate. This preconditioning adaptively rescales the gradient according to curvature magnitude, suppressing oscillations and accelerating convergence. The curvature matrix, $H_{\text{fuse}}$, thus serves as a dynamic metric guiding updates in both global and local subspaces.

The Newton-step style optimization further promotes smooth and stable decision boundaries. By integrating the embedding Hessians from both teachers, the student implicitly learns to maintain flatness in high-curvature regions

and robustness against strong augmentations. This contrasts with conventional teacher–student methods, which rely purely on first-order gradients. Through second-order guidance, TTN encourages the student to follow a curvature-informed learning path that achieves faster convergence and stronger generalization under limited-label conditions.

### 3.3. Training Objective

The overall training objective of TTN integrates both labeled and unlabeled supervision within a unified optimization framework. The student model, $S$, learns from two data sources: a labeled set, $\mathcal{D}_l = \{(x_l, y_l)\}$, and an unlabeled set, $\mathcal{D}_u = \{x_u\}$. The labeled data provide direct categorical supervision, while the unlabeled data contribute soft pseudo-label guidance, consistency regularization, and curvature information distilled from the two teachers.

The total training objective is defined in Eq. 6. Let $S(x)$ denote the prediction vector produced by the student model for input $x$, e.g., the softmax class-probability output. The supervised loss is computed as the standard cross-entropy on labeled samples:

$$\mathcal{L}_{\text{sup}} = \frac{1}{|\mathcal{D}_l|} \sum_{(x_l, y_l) \in \mathcal{D}_l} \text{CE}\big(S(x_l), y_l\big), \qquad (8)$$

and the unsupervised loss consists of two complementary parts: a pseudo-label alignment term and a prediction consistency regularization term. Specifically, it encourages the student prediction on strongly augmented inputs, $x_u^s$, to match the fused pseudo-label, $\hat{y}_u$, while maintaining prediction consistency across weak and strong augmentations:

$$\mathcal{L}_{\text{unsup}} = \frac{1}{|\mathcal{D}_u|} \sum_{x_u \in \mathcal{D}_u} \Big[ \text{CE}\big(S(x_u^s), \hat{y}_u\big) + \mathcal{L}_{\text{cons}} \Big], \qquad (9)$$

where the consistency regularization objective is defined as

$$\mathcal{L}_{\text{cons}} = \| S(x_u^s) - S(x_u^w) \|_2^2. \qquad (10)$$

The first term in Eq. 9 enforces semantic alignment between the student and the fused dual-teacher supervision, while the second term in Eq. 10 encourages stable predictions under different augmentations. This strategy ensures that the student remains robust to data augmentation and local perturbations without introducing additional optimization branches or hyperparameters.

As a whole, $\mathcal{L}_{\text{unsup}}$ captures both semantic alignment and prediction consistency. The pseudo-label alignment guides the student towards the teachers' fused supervision, while the consistency regularization maintains smooth predictions in the neighborhood of each unlabeled sample. Together with the Newton-step style update governed by $H_{\text{fuse}}$, our training objective forms a unified second-order learning framework that stabilizes optimization and enhances generalization under limited-label settings.

# 4. Theoretical Guarantee

We present a theoretical analysis of the proposed TTN framework to characterize how confidence-weighted dual-teacher fusion and curvature-guided optimization improve stability and convergence in SSL. Our analysis covers pseudo-label fusion in the probability space, curvature fusion in the parameter space, Newton-style optimization dynamics, and variance reduction induced by fused supervision. Together, the following theorems establish the geometric, optimization, and statistical foundations of the proposed framework.

To enable principled multi-teacher supervision, the fused pseudo-label must be confidence-aware, probabilistically grounded, and consistent with the geometry of predictive distributions. We formalize this requirement using a confidence-weighted KL-barycenter, which preserves distributional structure while adapting to teacher reliability.

**Definition 4.1** (Confidence-Weighted Teacher Outputs). Let $T_A$ and $T_B$ be two teacher models based on Eq. (1). For a given input $x$, they produce softmax probability distributions:

$$p_A = \hat{y}_u^{(A)}, \quad p_B = \hat{y}_u^{(B)}. \quad (11)$$

The confidence score of each teacher is defined as the maximum predicted class probability (see Eq. (2)):

$$\omega_A = \max_c p_A[c], \quad \omega_B = \max_c p_B[c]. \quad (12)$$

The normalized confidence weights are given by

$$\tilde{\omega}_A = \frac{\omega_A}{\omega_A + \omega_B}, \quad \tilde{\omega}_B = 1 - \tilde{\omega}_A. \quad (13)$$

**Theorem 4.2** (KL-barycenter of Two Distributions). *Under Definition 4.1, let us assume that $p_A$ and $p_B$ belong to the same exponential family. The KL-barycenter of $p_A$ and $p_B$ with weights $(\tilde{\omega}_A, \tilde{\omega}_B)$ is given by the normalized weighted geometric mean:*

$$\hat{y}_u[c] = \frac{p_A[c]^{\tilde{\omega}_A} p_B[c]^{\tilde{\omega}_B}}{\sum_k p_A[k]^{\tilde{\omega}_A} p_B[k]^{\tilde{\omega}_B}}. \quad (14)$$

*Moreover, this distribution is the unique minimizer of the weighted sum of KL divergences:*

$$\hat{y}_u = \arg\min_q \left( \tilde{\omega}_A \mathrm{KL}(q \mid p_A) + \tilde{\omega}_B \mathrm{KL}(q \mid p_B) \right). \quad (15)$$

Note that Eq. (4) and the expression in Theorem 4.2 are mathematically equivalent, differing only in notation. In Theorem 4.2, the KL-barycenter is written component-wise for two generic probability distributions, $p_A$ and $p_B$, yielding $q[c] \propto p_A[c]^{\tilde{\omega}_A} p_B[c]^{\tilde{\omega}_B}$. In Eq. (4), we instantiate this result by setting $p_A = \hat{y}_u^{(A)}$ and $p_B = \hat{y}_u^{(B)}$, and write the same normalized weighted geometric mean in vector form.

This result provides a principled justification for our pseudo-label fusion strategy. Unlike arithmetic averaging or heuristic ensembling, the proposed framework operates in the log-probability space and adapts to teacher confidence, producing a geometrically consistent and reliable supervision signal. By emphasizing agreement while attenuating uncertain predictions, this fusion directly improves training stability. A detailed proof of Theorem 4.2, based on properties of exponential families and Bregman divergences, is provided in Appendix A.1.

Building on this probabilistic fusion, we next extend the confidence-weighted aggregation from the prediction space to the optimization geometry by incorporating second-order information. Specifically, we study how local curvature estimates from multiple teachers can be fused into a single preconditioning metric that preserves positive definiteness and promotes smoother optimization dynamics.

**Assumption 4.3** (Teacher Curvature Regularity). Let $H_A$ and $H_B$ denote the local Hessian matrices in the embedding space induced by teacher models $T_A$ and $T_B$, respectively. We assume that $H_A$ and $H_B$ are symmetric positive definite matrices. The fused inverse Hessian is defined via a confidence-weighted harmonic mean (see Eq. (5)):

$$H_{\text{fuse}}^{-1} = \tilde{\omega}_A H_A^{-1} + \tilde{\omega}_B H_B^{-1}. \quad (16)$$

**Theorem 4.4** (Confidence-Weighted Hessian Fusion). *Under Assumption 4.3, the harmonic mean fusion corresponds to the optimal preconditioning matrix in the Riemannian geometry associated with the weighted KL divergence. The fused curvature satisfies the trace (*$\mathrm{Tr}$*) inequality:*

$$\mathrm{Tr}(H_{\text{fuse}}) \leq \tilde{\omega}_A \mathrm{Tr}(H_A) + \tilde{\omega}_B \mathrm{Tr}(H_B). \quad (17)$$

This result shows that confidence-weighted harmonic fusion yields a curvature matrix with reduced magnitude and improved conditioning compared to linear Hessian combinations, suppressing sharp directions in the loss landscape. When applied in Newton-style updates, the fused curvature guides the student along geometrically consistent directions from both teachers, leading to smoother and more stable optimization. A proof of Theorem 4.4 is provided in the Appendix A.2. We next analyze the optimization behavior of this curvature-guided update under a second-order loss approximation and show how fused-Hessian preconditioning improves convergence over first-order methods.

**Definition 4.5** (Local Quadratic Approximation). Let $\mathcal{L}(\theta)$ be a twice continuously differentiable loss function. Around a point $\theta$, we consider a small perturbation $d$ in the parameter space, where $d$ denotes a local update direction from $\theta$ to $\theta + d$. The second-order Taylor expansion of $\mathcal{L}$ is given by

$$\mathcal{L}(\theta + d) = \mathcal{L}(\theta) + g^\top d + \frac{1}{2} d^\top H d + o(\|d\|^2), \quad (18)$$

*Table 1.* Configuration of TTN. Both teachers MAE (ViT-B/16) and DINOv3 (ViT-B/14) are frozen during training, and the student is optimized using Newton-guided updates.

| Teacher A | Teacher B | Student | Params (M) | FLOPs |
|-----------|-----------|---------|------------|-------|
| MAE | DINOv3 | ViT-B | 90 | 64.8G |

where $g = \nabla \mathcal{L}(\theta)$ and $H = \nabla^2 \mathcal{L}(\theta)$ denote the gradient and Hessian at $\theta$, respectively.

**Theorem 4.6** (Local Linear Convergence of Fused Newton Update). *Under Definition 4.5, assume that the fused curvature matrix $H_{fuse}$ is symmetric positive definite and spectrally bounded with respect to the true Hessian $H$, that is, there exist constants $\mu, L > 0$, such that*

$$\mu I \preceq H_{fuse} \preceq LI. \tag{19}$$

*where $I$ is identity matrix. Consider the preconditioned update rule in Eq. (7):*

$$\theta_{t+1} = \theta_t - \eta H_{fuse}^{-1} g_t, \tag{20}$$

*where $g_t = \nabla \mathcal{L}(\theta_t)$ and the step size is chosen as $\eta = \frac{2}{\mu + L}$. Then, in a neighborhood of the optimum $\theta^*$, the iterations satisfy the linear convergence bound:*

$$\|\theta_{t+1} - \theta^*\| \leq \frac{\kappa - 1}{\kappa + 1} \|\theta_t - \theta^*\|, \tag{21}$$

*where $\kappa = \frac{L}{\mu}$ denotes the condition number of $H_{fuse}$.*

This result shows that curvature-guided updates based on the fused Hessian yield a contraction factor determined by the condition number of the preconditioner rather than that of the raw loss Hessian. Consequently, when $H_{\text{fuse}}$ provides a better-conditioned approximation of $H$, the resulting convergence rate is strictly faster than that of standard gradient descent, supporting the use of fused second-order information to accelerate and stabilize optimization. A detailed proof of Theorem 4.6, based on classical results from preconditioned gradient methods, is provided in Appendix A.3.

We next turn to the statistical effects of confidence-weighted pseudo-label fusion, and analyze how combining teacher predictions influences the variance of the resulting supervision signal under mild probabilistic assumptions.

**Definition 4.7** (Fused Pseudo-Label and Cross-Entropy Loss). Let $\hat{y}_u$ denote the confidence-weighted fused pseudo-label obtained from two teacher predictions, and let $y$ be the ground-truth label. The cross-entropy loss between the fused pseudo-label and the true label is defined as:

$$\text{CE}(\hat{y}_u, y) = -\sum_c y[c] \log \hat{y}_u[c]. \tag{22}$$

Let $S(x)$ denote the prediction vector produced by the student model for input $x$, and $x_u^w$ and $x_u^s$ represent the weakly

and strongly augmented versions of the same unlabeled input, respectively.

**Theorem 4.8** (Variance Reduction via Confidence-Weighted Fusion). *Assume that the prediction errors of the two teachers have finite second moments and are uncorrelated. Let $\epsilon_A = p_A - y$ and $\epsilon_B = p_B - y$ denote the prediction errors of the two teachers, with covariance matrices $\Sigma_A$ and $\Sigma_B$, respectively. Under the local linear fusion approximation, the fused prediction error is given by*

$$\epsilon_{\text{fuse}} = \tilde{\omega}_A \epsilon_A + \tilde{\omega}_B \epsilon_B, \tag{23}$$

*where $\tilde{\omega}_A, \tilde{\omega}_B \in [0, 1]$ and $\tilde{\omega}_A + \tilde{\omega}_B = 1$. Then the fused pseudo-label $\hat{y}_u$ satisfies the variance bound:*

$$\begin{aligned} \text{Tr}(\Sigma_{\text{fuse}}) &\leq \tilde{\omega}_A^2 \text{Tr}(\Sigma_A) + \tilde{\omega}_B^2 \text{Tr}(\Sigma_B) \\ &\leq \tilde{\omega}_A \text{Tr}(\Sigma_A) + \tilde{\omega}_B \text{Tr}(\Sigma_B). \end{aligned} \tag{24}$$

*Consequently,*

$$\text{Tr}(\Sigma_{\text{fuse}}) \leq \max\left\{\text{Tr}(\Sigma_A), \text{Tr}(\Sigma_B)\right\}. \tag{25}$$

*If the two teachers have identical error covariance, i.e., $\Sigma_A = \Sigma_B = \Sigma$, then*

$$\text{Tr}(\Sigma_{\text{fuse}}) \leq \text{Tr}(\Sigma), \tag{26}$$

*with strict reduction when both teachers receive non-zero weights and their errors are not perfectly correlated.*

This result shows that confidence-weighted fusion can reduce the total variance of the supervision signal, especially when the two teachers provide complementary and non-perfectly correlated errors. The fused pseudo-label is therefore more stable than supervision from a single noisy source in the local linear regime. This variance reduction complements the curvature-guided optimization of TTN, leading to more stable training under strong augmentations. A proof of Theorem 4.8 is provided in Appendix A.4.

Together, Theorems 4.2–4.8 establish a unified theoretical foundation for TTN. The framework achieves geometric consistency via KL-barycenter pseudo-label fusion, constructs a well-conditioned second-order geometry through harmonic curvature fusion, accelerates optimization with Newton-style updates, and stabilizes learning by reducing the total variance of the fused supervision signal. These results provide a principled alternative to heuristic ensembling and zero-order supervision under limited-label settings.

## 5. Experiments

**Datasets.** We evaluate TTN on four standard semi-supervised benchmarks: ImageNet (Deng et al., 2009), CIFAR-10 (Krizhevsky et al., 2009), SVHN (Netzer et al., 2011), and STL-10 (Coates et al., 2011). For ImageNet, we

Table 2. Top-1 accuracy (%, mean ± std) on ImageNet under 1%, 10%, and 25% labeled data settings.

| Model | Type | Method | 1% | 10% | 25% | Param. (M) |
|---|---|---|---|---|---|---|
| FixMatch (Sohn et al., 2020) | Consistency-based | Pseudo-labeling | 52.6±0.32 | 68.7±0.28 | 74.9±0.25 | 25.0 |
| UDA (Xie et al., 2020) | Consistency-based | Distribution alignment | 51.2±0.35 | 67.5±0.30 | 73.8±0.27 | 25.0 |
| FlexMatch (Zhang et al., 2021) | Confidence-aware | Adaptive threshold | 53.5±0.30 | 70.2±0.26 | 75.3±0.24 | 25.0 |
| Meta Pseudo Label (Pham et al., 2021) | Meta-learning | Meta pseudo-labeling | 55.0±0.29 | 71.8±0.24 | 76.4±0.22 | 28.0 |
| SimCLRv2+KD (Chen et al., 2020) | Self-supervised + KD | Distillation | 54.5±0.31 | 69.7±0.27 | 75.5±0.24 | 30.0 |
| Semi-ViT (ViT-B) (Cai et al., 2022) | Self-training | Self-labeled | 74.1±0.21 | 81.6±0.18 | 84.2±0.16 | 86.0 |
| Semi-ViT (ViT-L) (Cai et al., 2022) | Self-training | Self-labeled | 77.3±0.20 | 83.3±0.17 | 85.1±0.15 | 307.0 |
| Semi-ViT (ViT-H) (Cai et al., 2022) | Self-training | Self-labeled | 78.9±0.18 | 84.6±0.16 | 86.2±0.14 | 632.0 |
| REACT (ViT-L) (Liu et al., 2023) | Robust SSL | Distribution calibration | **81.6**±0.15 | 85.1±0.13 | 86.8±0.12 | 307.0 |
| SemiFormer (Weng et al., 2022) | Semi-supervised ViT | Confidence teacher | 75.8±0.22 | 82.1±0.19 | 84.5±0.17 | 86.0 |
| DINO (ViT-L) (Caron et al., 2021) | Self-supervised | Linear head | 78.1±0.19 | 82.9±0.17 | 84.9±0.15 | 307.0 |
| Semi-SST (ViT-H) (Zhao et al., 2025) | Semi-supervised ViT | Self-adaptive thresholding | 80.7±0.25 | 84.9±0.13 | – | 632.0 |
| Co-Training (Rothenberger & Diochnos, 2023) | Co-training | Two-view mutual labeling | 80.1±0.17 | 85.1±0.15 | – | 45.0 |
| MCT (Rothenberger & Diochnos, 2023) | Meta Co-training | Meta feedback | 80.7±0.16 | 85.8±0.14 | – | 47.0 |
| TRiCo (He et al., 2025a) | Game-theoretic SSL | TRiCo-Training | 81.2±0.14 | **85.9**±0.12 | 88.3±0.11 | 95.0 |
| **TTN (ours)** | Dual-teacher SSL | Newton-guided update | 81.4±0.13 | 85.7±0.12 | **88.5**±0.10 | 90.0 |

adopt the widely used low-label protocol and train on 1%, 10%, and 25% labeled subsets while treating the remaining images as unlabeled. We use the full labeled set together with all unlabeled data for training. All images are resized to 32×32 for CIFAR-10 and SVHN, and 96×96 for STL-10, while ImageNet inputs are resized to 224×224 with standard random cropping and flipping.

**Implementation details.** All experiments are implemented in PyTorch. Both teachers (MAE (He et al., 2022) and DI-NOv3 (Oquab et al., 2023)) are self-supervised and kept frozen. The student uses a ViT-B backbone initialized from ImageNet supervised pretraining (Zhai et al., 2022). Training employs AdamW (Loshchilov & Hutter, 2017) with an initial learning rate of $3 \times 10^{-4}$, weight decay $5 \times 10^{-4}$, and cosine scheduling. We train for 200k steps on ImageNet and 100k steps on CIFAR-10, SVHN, and STL-10. Student updates follow Newton-guided optimization (see Eq. 7) with diagonal fused-Hessian preconditioning (Nocedal, 2006; Martens et al., 2010; Anil et al., 2021). An EMA with decay 0.995 is used for evaluation (Tarvainen & Valpola, 2017), and weak/strong augmentations follow Fix-Match (Sohn et al., 2020). All experiments are conducted on a single NVIDIA A5000 GPU.

Table 1 summarizes the configuration of TTN, where FLOPs denote the total forward-pass computational cost for a batch size of 64 and a 224×224 input resolution as measured on an NVIDIA RTX 5090, which is the hardware used exclusively for computational profiling.

### 5.1. Results and Analysis

Table 2 compares performance on ImageNet with 1%, 10%, and 25% labeled data. TTN achieves accuracy comparable to TRiCo with fewer trainable parameters, demonstrating the efficiency of its dual-teacher Newton-guided design. Under the 1% labeled data setting, TTN reaches 81.4%,

Table 3. Top-1 accuracy (%, mean ± std) on CIFAR-10, SVHN, and STL-10. CIFAR-10 uses 4k labeled samples and SVHN uses 1k labeled samples. For STL-10, we follow the standard SSL protocol and use the official 5k labeled training split together with the official 100k unlabeled split; the unlabeled data are provided by STL-10 and are not taken from an external dataset.

| Method | CIFAR-10 (4k) | SVHN (1k) | STL-10 (5k + 100k) |
|---|---|---|---|
| FixMatch | 94.3±0.24 | 92.1±0.26 | 89.5±0.28 |
| UDA | 93.4±0.27 | 91.2±0.29 | 88.6±0.30 |
| FlexMatch | 94.9±0.22 | 92.7±0.24 | 90.1±0.26 |
| Meta Pseudo Label | 95.1±0.21 | 93.5±0.22 | 90.6±0.24 |
| Semi-ViT (ViT-B) | 94.6±0.23 | 91.7±0.25 | 89.0±0.27 |
| Semi-ViM-Base | 96.0±0.18 | 93.9±0.19 | 92.3±0.20 |
| TRiCo | 96.3±0.16 | **94.2**±0.15 | 93.1±0.17 |
| **TTN (ours)** | **97.5**±0.12 | 93.6±0.16 | **94.4**±0.14 |

matching REACT and slightly surpassing TRiCo (81.2%). As the label ratio increases, TTN remains competitive with a 85.7% and 88.5% accuracy for the 10% and 25% labeled data settings, respectively, outperforming recent methods; i.e., Semi-ViT, MCT, and Meta Pseudo Label. Compared to consistency-based and self-supervised baselines, TTN more effectively exploits unlabeled data via confidence-weighted fusion and curvature-aware optimization, offering a lightweight yet robust alternative without meta-learning or adversarial tuning.

Across small- and medium-scale benchmarks, TTN consistently outperforms recent SSL methods. As shown in Table 3, TTN achieves strong generalization under varying data complexity and label scarcity. On CIFAR-10 (4k labels), TTN attains 97.5% accuracy, surpassing TRiCo and other strong baselines. On SVHN, TTN remains competitive compared to TRiCo despite severe label scarcity. On STL-10, TTN achieves the highest accuracy of 94.4%. These results demonstrate that confidence-weighted dual-teacher fusion and curvature-guided optimization improve pseudo-label reliability and enable stable cross-domain adaptation without increasing model complexity.

Figure 3 illustrates the training dynamics of TTN, in di-

*Table 4.* Ablation study on CIFAR-10 with 4k labeled samples. "Conf. Fusion" denotes confidence-weighted pseudo-label fusion. "Newton" denotes the Newton-guided update. "Cons." denotes the augmentation consistency loss $L_{cons}$ in Eq. (10) and not teacher-teacher consistency. For single-teacher variants, the same $L_{cons}$ is retained, while the dual-teacher fusion branch is removed. For single-teacher variants, "–" in the "Conf. Fusion" column means that no dual-teacher fusion is applied; the pseudo-label and curvature estimate are directly obtained from the single teacher.

| VARIANT | DUAL TEACHER | CONF. FUSION | NEWTON | $\mathcal{L}_{\mathbf{cons}}$ | ACC. (%) |
|---|---|---|---|---|---|
| TTN (FULL) | ✓ | ✓ | ✓ | ✓ | **97.5**±0.12 |
| W/O NEWTON UPDATE | ✓ | ✓ | ✗ | ✓ | 89.8±0.28 |
| W/O CONF. FUSION | ✓ | ✗ | ✓ | ✓ | 91.2±0.24 |
| SINGLE-TEACHER (MAE ONLY) | ✗ | – | ✓ | ✓ | – |
| SINGLE-TEACHER (DINOv3 ONLY) | ✗ | – | ✓ | ✓ | 91.0±0.25 |
| W/O CONSISTENCY ($\mathcal{L}_{cons}$) | ✓ | ✓ | ✓ | ✗ | 92.3±0.21 |
| EQUAL HESSIAN FUSION | ✓ | ✓ | ✓ | ✓ | 92.0±0.22 |
| NO CURVATURE/NEWTON-STEP (Identity $H{=}I$) | ✓ | ✓ | ✗ | ✓ | 90.8±0.26 |

*Table 5.* Convergence speed comparison under the same training setup. Time / Iter. is measured on a single NVIDIA RTX 5090 using the same data pipeline, augmentation policy, optimizer setting, and batch composition. N/R means that the target accuracy was not reached.

| Dataset | Method | Final Acc. | Target 1 | Step | Time | Target 2 | Step | Time | Time / Iter. |
|---|---|---|---|---|---|---|---|---|---|
| CIFAR-100 | TTN | 81.4 | 75% | 2.8k | 2.9 ks | 80% | 6.0k | 6.3 ks | 0.992 s |
| CIFAR-100 | w/o Newton | 80.0 | 75% | 4.0k | 4.1 ks | 80% | 11.0k | 10.5 ks | 0.973 s |
| CIFAR-10 | TTN | 97.5 | 85% | 2.2k | 2178 s | 93% | 6.5k | 6435 s | 0.987 s |
| CIFAR-10 | w/o Newton | 89.8 | 85% | 4.8k | 4656 s | 93% | N/R | N/R | 0.969 s |

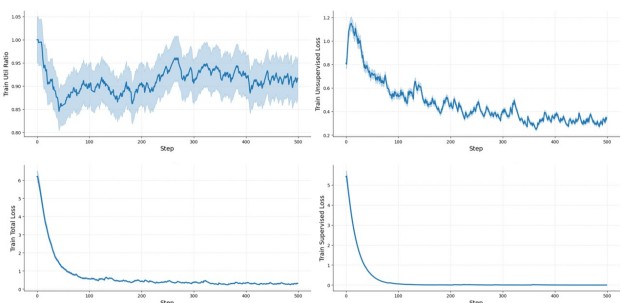

*Figure 3.* **Training dynamics of TTN.** TTN shows smoother loss curves and more stable pseudo-label utilization compared with conventional SSL training.

rect contrast to the oscillatory behavior observed in conventional SSL frameworks (Figure 1). TTN maintains a stable pseudo-label utilization ratio of 0.9, reflecting that pseudo-label adoption is adaptively moderated by the confidence-weighted dual-teacher fusion rather than determined by a fixed threshold. Both unsupervised and total losses decrease smoothly with minimal variance, indicating that the Newton-Step style update effectively mitigates fluctuations caused by non-flat decision boundaries. The supervised loss converges rapidly and remains nearly constant, showing that the optimization is dominated by reliable unlabeled guidance while preserving stable performance on labeled data.

# 6. Ablation study

Figure 4 illustrates backbone performance and curvature evolution during training. Transformer-based models (ViT-S and ViT-B) consistently outperform convolutional backbones under the 1% and 25% ImageNet labeled data settings, with the gap widening at higher label ratios. The curvature plots show that the single-teacher baseline suffers from frequent curvature spikes, particularly in the unsupervised loss, indicating unstable optimization. In contrast, TTN exhibits smooth and monotonic curvature decay, demonstrating that Newton-guided preconditioning effectively stabilizes training. The reduced curvature variance aligns with improved accuracy, confirming that curvature regularization directly enhances generalization and convergence stability.

Table 4 presents an ablation study of TTN, showing that removing any component leads to a clear performance drop. The Newton-step update is the most critical component: removing it decreases accuracy from 97.5% to 89.8%, highlighting the importance of second-order preconditioning for stable optimization. Disabling confidence-weighted dual-teacher fusion or replacing harmonic Hessian fusion with arithmetic averaging also leads to clear degradation, confirming the need for adaptive teacher reliability weighting. For the single-teacher variants, we use only one teacher model rather than duplicating the same teacher twice. Thus, no dual-teacher pseudo-label fusion or dual-teacher curvature fusion is performed in these variants; the pseudo-label and curvature estimate are taken directly from the single teacher. The performance drop of the single-teacher setting

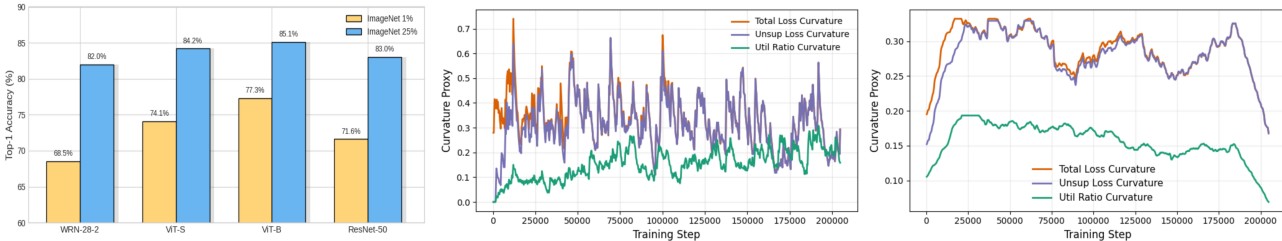

*Figure 4.* **Visualization of curvature dynamics and backbone performance.** Left: Top-1 accuracy of different backbones (WRN-28-2, ViT-S, ViT-B, ResNet-50) under the 1% and 25% ImageNet labeled data settings. Middle: curvature dynamics of the **Single-Teacher (DINOv3 only)** baseline, measured by the trace norm of the Hessian proxy for the total, unsupervised, and utilization losses. Right: curvature dynamics of **TTN** under the same training setting and using the same curvature proxy. The middle and right plots use the same x-axis, loss components, and curvature measure to enable a direct comparison between the single-teacher baseline and TTN. Compared with the single-teacher baseline, TTN maintains lower and smoother curvature throughout training, indicating that the Newton-guided dual-teacher update suppresses sharp curvature fluctuations and improves optimization stability.

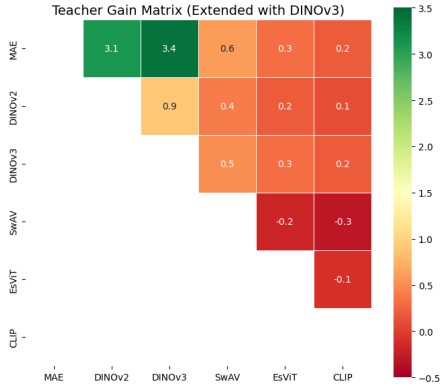

*Figure 5.* **Teacher gain matrix on ImageNet (25% labeled data setting)**. The matrix shows top-1 accuracy gains (%) for different self-supervised teacher pairs in TTN. Green cells indicate positive synergy, while red cells denote redundancy.

therefore reflects the benefit of complementary teacher supervision rather than an artifact of reusing one teacher in the fusion module. Removing the consistency regularization term also reduces performance, showing that augmentation consistency remains useful alongside dual-teacher supervision. Overall, TTN's gains arise from the coordinated effect of complementary dual-teacher fusion, Newton-guided optimization, and consistency regularization.

Figure 5 shows the pairwise gain matrix among six representative self-supervised teachers. MAE and DINOv3 form the most effective combination, achieving a 3.4% improvement over the baseline, followed by the MAE–DINOv2 pair (3.1%). These results indicate that heterogeneous representation priors—MAE focusing on low-level structure and DINOv3 capturing high-level semantics—yield the strongest complementary supervision. In contrast, pairs involving SwAV or EsViT show limited or even negative synergy, suggesting that excessive similarity in inductive biases leads to redundant guidance for the student. These results suggest that TTN is not restricted to the MAE–DINOv3 pair. Although DINOv3 is a strong teacher, teacher strength alone does not fully explain the gain, since different pairs involv-

ing strong self-supervised models show different levels of improvement. The benefit mainly comes from complementary representation priors between teachers: reconstruction-oriented teachers provide local structural cues, while contrastive or distillation-based teachers provide more global semantic cues.

### 6.1. Convergence Speed Analysis

Table 5 further validates the practical benefit of the Newton-guided update. Under the same hardware, data pipeline, augmentation policy, optimizer setting, and batch composition, TTN reaches the same target accuracy in fewer training steps while maintaining nearly unchanged per-iteration cost. This indicates that the acceleration mainly comes from better-conditioned optimization rather than from additional computation. The dominant overhead is caused by the frozen dual-teacher forward passes, whereas the diagonal Newton preconditioning itself is lightweight.

## 7. Conclusion and Limitations

TTN offers a novel perspective on SSL by coupling dual self-supervised teachers with Newton-guided second-order optimization. This design transforms the teacher–student interaction from categorical imitation to geometry-consistent adaptation, enabling smoother convergence and stronger generalization under limited labels. While the current formulation relies on diagonal Hessian approximation and frozen teacher encoders, future work can explore more expressive curvature modeling and jointly adaptive teachers to further enhance second-order guidance.

A practical limitation of TTN is its training-time overhead from dual frozen teacher forward passes. This overhead is not required at test time, since only the trained student is used for inference. Moreover, the Newton-guided component itself is lightweight: it uses diagonal curvature estimates and element-wise preconditioning, without full Hessian construction or second-order backpropagation.

## Impact Statement

This paper aims to advance SSL and optimization for visual recognition. TTN may reduce annotation cost by improving learning from limited labeled data. The method is intended for general machine learning research and does not target sensitive applications directly. Like other SSL methods, its performance can be affected by biased or noisy unlabeled data. We do not identify additional societal risks beyond those commonly associated with visual recognition models.

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

# A. Proof of the Theorems

## A.1. Proof of Theorem 4.2

*Proof.* We characterize the solution of the weighted Kullback–Leibler minimization problem in closed form.

Let $p$ be a distribution in an exponential family. It admits the representation

$$p[c] = \exp(\theta_c - A(\theta)), \tag{27}$$

where $\theta$ denotes the natural parameter vector and $A(\theta)$ is the log-partition function.

For exponential family distributions, the KL divergence between $q$ and $p$ can be written as

$$\begin{aligned}
\mathrm{KL}(q \mid p) &= \mathbb{E}_q[\log q] - \mathbb{E}_q[\log p] \\
&= \mathbb{E}_q[\log q] - \theta^\top \mathbb{E}_q[\phi] + A(\theta),
\end{aligned} \tag{28}$$

where $\phi$ denotes the sufficient statistic.

Consider the optimization problem in Eq. (15)

$$\min_{q \in \Delta} \left( \tilde{\omega}_A \mathrm{KL}(q \mid p_A) + \tilde{\omega}_B \mathrm{KL}(q \mid p_B) \right), \tag{29}$$

where $\Delta = \{q \mid q[c] \geq 0, \sum_c q[c] = 1\}$ is the probability simplex.

Introducing a Lagrange multiplier $\lambda$ for the normalization constraint, the Lagrangian is

$$L(q, \lambda) = \tilde{\omega}_A \sum_c q[c] \big( \log q[c] - \log p_A[c] \big) + \tilde{\omega}_B \sum_c q[c] \big( \log q[c] - \log p_B[c] \big) + \lambda \left( \sum_c q[c] - 1 \right). \tag{30}$$

Taking partial derivatives with respect to $q[c]$ and imposing stationarity yields

$$\frac{\partial L}{\partial q[c]} = \tilde{\omega}_A \big( \log q[c] + 1 - \log p_A[c] \big) + \tilde{\omega}_B \big( \log q[c] + 1 - \log p_B[c] \big) + \lambda = 0. \tag{31}$$

Rearranging terms gives

$$(\tilde{\omega}_A + \tilde{\omega}_B) \log q[c] = \tilde{\omega}_A \log p_A[c] + \tilde{\omega}_B \log p_B[c] - (\tilde{\omega}_A + \tilde{\omega}_B) - \lambda. \tag{32}$$

Since $\tilde{\omega}_A + \tilde{\omega}_B = 1$, this reduces to

$$\log q[c] = \tilde{\omega}_A \log p_A[c] + \tilde{\omega}_B \log p_B[c] + C, \tag{33}$$

where $C = -1 - \lambda$ is a normalization constant independent of $c$.

Exponentiating both sides of Eq. (33) yields:

$$q[c] \propto \exp \left( \tilde{\omega}_A \log p_A[c] + \tilde{\omega}_B \log p_B[c] \right) = p_A[c]^{\tilde{\omega}_A} p_B[c]^{\tilde{\omega}_B}. \tag{34}$$

Normalizing over $c$ completes the proof. $\square$

## A.2. Proof of Theorem 4.4

*Proof.* We proceed in three steps.

First, under assumption 4.3, by a classical matrix inequality for symmetric positive-definite matrices, for any $A \succ 0$, $B \succ 0$, and any $t \in [0, 1]$, the harmonic mean satisfies:

$$(tA^{-1} + (1-t)B^{-1})^{-1} \preceq tA + (1-t)B. \tag{35}$$

Taking $t = \tilde{\omega}_A$ and using $\tilde{\omega}_B = 1 - \tilde{\omega}_A$, we obtain:

$$H_{\text{fuse}} \preceq \tilde{\omega}_A H_A + \tilde{\omega}_B H_B. \tag{36}$$

Since the cone of symmetric positive-definite matrices is closed under convex combinations, it follows that $H_{\text{fuse}} \succ 0$.

Second, applying the trace operator to both sides of the Loewner inequality and using its monotonicity on the positive semidefinite cone yields:

$$\text{Tr}(H_{\text{fuse}}) \leq \tilde{\omega}_A \text{Tr}(H_A) + \tilde{\omega}_B \text{Tr}(H_B). \tag{37}$$

Hence, the fused curvature matrix admits an upper bound in trace by the confidence-weighted arithmetic mean of the individual curvatures.

Third, consider an exponential family $\{p_\theta\}$ parameterized by $\theta$. The Fisher information matrix satisfies:

$$F_\theta = -\mathbb{E}_{p_\theta} \left[ \nabla^2_\theta \log p_\theta \right]. \tag{38}$$

For $\theta$ in a neighborhood of $\theta_0$, the KL divergence admits the second-order expansion

$$\text{KL}(p_{\theta_0} \mid p_\theta) = \frac{1}{2}(\theta - \theta_0)^\top F_{\theta_0}(\theta - \theta_0) + o(\|\theta - \theta_0\|^2). \tag{39}$$

Let $F_A$ and $F_B$ denote the Fisher information matrices associated with the two teachers. Under this local quadratic approximation, harmonic mean fusion of inverse Hessians corresponds to a weighted averaging of local Riemannian metrics induced by $F_A$ and $F_B$ in the KL geometry. The resulting preconditioner defines a single metric that jointly upper-bounds the corresponding quadratic forms induced by the individual teachers.

$\square$

### A.3. proof of Theorem 4.6

*Proof.* Let $\mathcal{L} : \mathbb{R}^d \to \mathbb{R}$ be a twice continuously differentiable function that is $\mu$-strongly convex and $L$-smooth. Its Hessian $H(\theta)$ satisfies:

$$\mu I \preceq H(\theta) \preceq LI \tag{40}$$

for all $\theta$ in a neighborhood of the minimizer $\theta^*$, where $I$ is the identity matrix. Let $P = H_{\text{fuse}}^{-1}$ be a symmetric positive-definite preconditioning matrix. The update rule is:

$$\theta_{t+1} = \theta_t - \eta P g_t, \tag{41}$$

where $g_t = \nabla \mathcal{L}(\theta_t)$.

By $L$-smoothness of $\mathcal{L}$, for any update direction $d$, the following inequality holds:

$$\mathcal{L}(\theta_t + d) \leq \mathcal{L}(\theta_t) + g_t^\top d + \frac{L}{2}\|d\|^2. \tag{42}$$

Substituting $d = -\eta P g_t$ into Eq. (42) yields:

$$\mathcal{L}(\theta_{t+1}) - \mathcal{L}(\theta_t) \leq -\eta g_t^\top P g_t + \frac{L\eta^2}{2} g_t^\top P^2 g_t. \tag{43}$$

Define the $P$-weighted norm $\|x\|_P^2 = x^\top P x$. Using the spectral bounds of $P$, there exist constants $\lambda_{\min}(P)$ and $\lambda_{\max}(P)$ such that:

$$\lambda_{\min}(P)\|g_t\|^2 \leq \|g_t\|_P^2 \leq \lambda_{\max}(P)\|g_t\|^2, \tag{44}$$

and:

$$\|g_t\|_{P^2}^2 \leq \lambda_{\max}(P)\|g_t\|_P^2. \tag{45}$$

Therefore, Eq. (43) becomes:

$$\mathcal{L}(\theta_{t+1}) - \mathcal{L}(\theta_t) \leq -\eta\|g_t\|_P^2 + \frac{L\eta^2 \lambda_{\max}(P)}{2}\|g_t\|_P^2. \tag{46}$$

Choosing the step size $\eta = \frac{1}{L\lambda_{\max}(P)}$ gives:

$$\mathcal{L}(\theta_{t+1}) - \mathcal{L}(\theta_t) \leq -\frac{1}{2L\lambda_{\max}(P)} \|g_t\|_P^2. \tag{47}$$

By $\mu$-strong convexity, the gradient satisfies:

$$\|g_t\|^2 \geq 2\mu \left(\mathcal{L}(\theta_t) - \mathcal{L}(\theta^*)\right). \tag{48}$$

Combining inequalities (44) (45) (47) and (48) yields a linear convergence bound:

$$\mathcal{L}(\theta_{t+1}) - \mathcal{L}(\theta^*) \leq \left(1 - \frac{\mu\lambda_{\min}(P)}{L\lambda_{\max}(P)}\right) \left(\mathcal{L}(\theta_t) - \mathcal{L}(\theta^*)\right). \tag{49}$$

Noting that $\kappa(P) = \frac{\lambda_{\max}(P)}{\lambda_{\min}(P)}$ is the condition number of the preconditioner, the contraction factor depends inversely on $\kappa(P)$. This establishes linear convergence of the preconditioned update and completes the proof. $\square$

### A.4. Proof of Theorem 4.8

*Proof.* Let $y$ denote the ground-truth label, and define the prediction errors of the two teachers as:

$$\varepsilon_A = p_A - y, \quad \varepsilon_B = p_B - y. \tag{50}$$

Assume that $\mathbb{E}[\varepsilon_A] = 0$, $\mathbb{E}[\varepsilon_B] = 0$, and that $\varepsilon_A$ and $\varepsilon_B$ are uncorrelated random vectors with covariance matrices

$$\Sigma_A = \mathbb{E}[\varepsilon_A \varepsilon_A^\top], \quad \Sigma_B = \mathbb{E}[\varepsilon_B \varepsilon_B^\top]. \tag{51}$$

The fused prediction error is given by the confidence-weighted combination:

$$\varepsilon_{\text{fuse}} = \tilde{\omega}_A \varepsilon_A + \tilde{\omega}_B \varepsilon_B. \tag{52}$$

Since $\varepsilon_A$ and $\varepsilon_B$ are uncorrelated, the covariance matrix of $\varepsilon_{\text{fuse}}$ is:

$$\Sigma_{\text{fuse}} = \mathbb{E}[\varepsilon_{\text{fuse}} \varepsilon_{\text{fuse}}^\top] = \tilde{\omega}_A^2 \Sigma_A + \tilde{\omega}_B^2 \Sigma_B. \tag{53}$$

Taking the trace on both sides yields:

$$\text{Tr}(\Sigma_{\text{fuse}}) = \tilde{\omega}_A^2 \text{Tr}(\Sigma_A) + \tilde{\omega}_B^2 \text{Tr}(\Sigma_B). \tag{54}$$

Since $\tilde{\omega}_A^2 \leq \tilde{\omega}_A$ and $\tilde{\omega}_B^2 \leq \tilde{\omega}_B$ for $\tilde{\omega}_A, \tilde{\omega}_B \in [0, 1]$ with $\tilde{\omega}_A + \tilde{\omega}_B = 1$, it follows that:

$$\begin{aligned}
\text{Tr}(\Sigma_{\text{fuse}}) &\leq \tilde{\omega}_A \text{Tr}(\Sigma_A) + \tilde{\omega}_B \text{Tr}(\Sigma_B) \\
&\leq \max\left(\text{Tr}(\Sigma_A), \text{Tr}(\Sigma_B)\right).
\end{aligned} \tag{55}$$

Moreover, if $\tilde{\omega}_A, \tilde{\omega}_B \in [0, 1]$ and $\Sigma_A \neq \Sigma_B$, then the inequality is strict, implying a strict reduction in total variance. This establishes that confidence-weighted fusion reduces the prediction variance relative to either individual teacher and completes the proof. $\square$

## B. Practical Implementation of Newton-Guided Update

This section details the practical realization of the proposed Newton-guided update in TTN, bridging the gap between the theoretical formulation in Sec. 4 and its efficient implementation. We emphasize that TTN does not compute or invert full Hessian matrices; instead, it adopts a lightweight diagonal approximation derived from frozen teacher embedding geometry, resulting in negligible computational overhead.

**Diagonal curvature approximation.** Although the analysis in Sec. 4 is presented for general symmetric positive-definite curvature matrices, in practice we approximate the fused curvature $H_{\text{fuse}}$ using a diagonal estimator. This choice follows standard practice in scalable second-order optimization for deep networks and preserves the stabilizing effect of curvature-aware preconditioning.

For each teacher $T \in \{T_A, T_B\}$, let $z_T(x_u^w)$ denote the embedding of the weakly augmented unlabeled input $x_u^w$. We estimate a Fisher-style diagonal curvature proxy in the teacher embedding space:

$$\text{diag}(H_T) \approx \mathbb{E}_{x_u \sim \mathcal{D}_u} \left[ (\nabla_{z_T} \log p_T(x_u^w))^2 \right], \tag{56}$$

where $p_T(x_u^w)$ denotes the softmax output of teacher $T$, and the square is taken element-wise. This estimator captures local sensitivity of teacher predictions and is inexpensive to compute, as all teacher networks are frozen.

**Confidence-weighted curvature fusion.** Following the same confidence-weighting principle used for pseudo-label fusion, the diagonal curvature estimates from the two teachers are combined via a harmonic mean. Let $\tilde{\omega}_A$ and $\tilde{\omega}_B$ denote the normalized confidence weights defined in Eq. (13). The fused diagonal curvature is computed as

$$\text{diag}(H_{\text{fuse}}) = \left( \tilde{\omega}_A \, \text{diag}(H_A)^{-1} + \tilde{\omega}_B \, \text{diag}(H_B)^{-1} \right)^{-1}, \tag{57}$$

where inversion and summation are applied element-wise. This fusion suppresses ill-conditioned or unreliable curvature directions contributed by low-confidence teachers, yielding a well-conditioned preconditioner in practice.

**Newton-guided student update.** Given the fused diagonal curvature, the student update in Eq. (7) reduces to an element-wise preconditioned gradient step:

$$\theta_s \leftarrow \theta_s - \eta \, \text{diag}(H_{\text{fuse}})^{-1} \odot \nabla_{\theta_s} L_{\text{total}}, \tag{58}$$

where $\odot$ denotes element-wise multiplication. Importantly, this update does not require second-order backpropagation, full Hessian construction, or matrix inversion.

Algorithm 1 summarizes the practical implementation of the Newton-guided update used in TTN, highlighting that only diagonal curvature estimation and element-wise preconditioning are required.

---

**Algorithm 1** Practical Newton-Guided Student Update (TTN)

---

**Require:** Unlabeled batch $x_u^w$, student parameters $\theta_s$, teachers $T_A, T_B$
**Ensure:** Updated student parameters $\theta_s$
 1: Compute teacher predictions and confidence weights $\hat{y}_u^{(A)}, \hat{y}_u^{(B)}, \tilde{\omega}_A, \tilde{\omega}_B$
 2: Estimate diagonal curvature $\text{diag}(H_A), \text{diag}(H_B)$ from teacher embeddings
 3: Fuse curvature via harmonic mean:

$$\text{diag}(H_{\text{fuse}}) \leftarrow \left( \tilde{\omega}_A \text{diag}(H_A)^{-1} + \tilde{\omega}_B \text{diag}(H_B)^{-1} \right)^{-1}$$

 4: Compute student gradient $g \leftarrow \nabla_{\theta_s} L_{\text{total}}$
 5: Update student parameters:

$$\theta_s \leftarrow \theta_s - \eta \, \text{diag}(H_{\text{fuse}})^{-1} \odot g$$

---

**Computational overhead.** All curvature estimation and fusion operations scale linearly with the embedding dimension and are computed from frozen teacher networks. Consequently, the proposed Newton-guided update introduces negligible overhead compared to standard first-order optimizers such as AdamW, while providing significantly improved optimization stability.

## C. Computational Complexity Analysis

This section analyzes the computational overhead introduced by TTN and addresses the practicality of the proposed framework. Although TTN incorporates dual teachers and curvature-guided optimization, its additional cost remains modest due to frozen teacher networks and lightweight diagonal approximations.

TTN employs two self-supervised teachers, MAE and DINOv3, to generate pseudo-labels and curvature cues for unlabeled samples. Both teachers are frozen throughout training and are only used for forward inference. As a result, the dual-teacher design introduces a constant-factor increase in computation corresponding to additional forward passes, without any gradient backpropagation or parameter updates for the teachers.

The Newton-guided update relies on diagonal curvature estimation in the teacher embedding space. This estimation involves element-wise operations on teacher embeddings or prediction gradients and scales linearly with the embedding dimension $d$, i.e., $\mathcal{O}(d)$. No full Hessian construction, second-order backpropagation, or matrix storage is required. Consequently, the overhead of curvature estimation is negligible compared to the cost of network forward and backward passes.

Curvature fusion and preconditioning are implemented using element-wise arithmetic operations, including inversion and weighted summation of diagonal terms. These operations are independent of the network depth and parameter count, and therefore do not change the asymptotic complexity of training. The resulting preconditioned update has the same order of computational complexity as standard first-order optimization methods.

Table 6 summarizes the additional computational cost of TTN relative to a standard single-teacher, first-order SSL framework. Importantly, TTN introduces no additional trainable parameters and does not alter the asymptotic training complexity.

*Table 6.* Additional computational overhead of TTN relative to standard first-order SSL methods.

| Component | Extra Cost |
|---|---|
| Dual teacher forward passes | $\sim 1.9\times$ forward computation |
| Diagonal curvature estimation | $\mathcal{O}(d)$ (negligible) |
| Curvature inversion and fusion | Element-wise operations |
| Second-order backpropagation | Not required |
| Additional trainable parameters | None |

