# OpenReview forum: "Newton-coupled Dual-Teacher Semi-supervised Learning Framework"
_ICML.cc/2026/Conference — ICML 2026 regular_

### Official Review · Reviewer_R5Ez · 2026-03-06

**Soundness:** 4
**Presentation:** 4
**Significance:** 3
**Originality:** 3
**Overall Recommendation:** 5
**Confidence:** 4

**Summary:**

The paper studies limitations of existing semi-supervised learning methods based on the teacher–student framework. In most current approaches, a single teacher provides pseudo-labels that are used to train the student model. This form of supervision is essentially “zero-order,” since the student only receives class predictions or confidence scores. As a result, the training signal contains little information about the local structure of the representation space or the shape of the loss landscape. The authors argue that this can make optimization unstable and may limit generalization when the amount of labeled data is small.

To address this issue, the paper proposes a method called Two-Teachers Newton-guided Learning (TTN). The main idea is to use two frozen self-supervised teacher models with different representational strengths. In particular, the method uses MAE to capture fine-grained, patch-level visual information and DINOv3 to capture higher-level semantic structure. For unlabeled examples, the predictions of the two teachers are combined to produce a single supervision signal for the student. The pseudo-labels are fused using a confidence-weighted combination based on the Kullback–Leibler divergence. In addition, the method incorporates curvature information derived from the teachers’ embeddings, which is used to construct a diagonal approximation of the inverse Hessian. This approximation serves as a preconditioner during training, allowing the student’s parameters to be updated in a way that resembles a Newton step rather than standard gradient descent.

The authors provide several theoretical results supporting the design of the method.  The method is evaluated on several standard semi-supervised learning benchmarks, including ImageNet, CIFAR-10, SVHN, and STL-10.

**Compliance With Llm Reviewing Policy:**

Affirmed.

**Final Justification:**

Having read the authors' responses to my comments and other reviewers' comments, I remain positive about it and I think it should be accepted.

**Key Questions For Authors:**

No questions.

**Limitations:**

The main limitation of the paper seems to me to be the computational overhead of the dual-teacher setup.

**Strengths And Weaknesses:**

Strengths:

— Technically sound submission, including well-supported claims by both experimental and theoretical analysis.

— Superior optimization stability: The paper overcomes one of the known issues of traditional SSL methods, which often suffer from "oscillations" (the model gets confused by noisy pseudo-labels and its loss values bounce around wildly).

— Parameter efficiency: The method tends to be more parameter-efficient than other SOTA-techniques.

Weaknesses:

— Inference "Teacher" overhead: While the teachers are frozen, the method still needs to run a forward pass through both MAE and DINOv3 for every unlabeled image. This adds roughly 1.9x more forward computation compared to a single teacher approach. This could be an issue in industrial-scale applications.

— The method achieves state-of-the-art results and typically outperforms other approaches, though the performance gains are incremental rather than a significant departure.

---

> ### Author Rebuttal · Authors · 2026-03-30
>
> We thank the reviewer for their professional review and the positive feedback.
>
> ***Inference "Teacher" overhead***
>
> The concern is valid, but it should be stated more precisely as a training-time overhead rather than a mandatory test-time inference overhead. In TTN, both teachers are frozen and only used for forward passes during training; the Newton component itself is lightweight and does not introduce second-order backpropagation or extra trainable parameters. Thus, the main additional cost comes from dual-teacher forward passes, exactly as our Table 5 (in the Appendix) reports. The practical severity of this overhead further depends on the teacher scale used.
>
> ***Incremental performance gains***
>
> We agree that the improvements are not a dramatic departure in absolute accuracy. However, in modern SSL benchmarks, especially under extremely limited-label settings, improvements are often inherently incremental. In this context, we believe the main value of TTN is not only the accuracy gain itself, but also the new second-order supervision paradigm and the consistently improved stability and robustness it brings across settings.

---

> > ### Author Rebuttal · Reviewer_R5Ez · 2026-04-02
> >
> > I understand that's why my score was already positive. Thank you for your responses!

---

> > > ### Author Response · Authors · 2026-04-03
> > >
> > > Thank you for your kind acknowledgement and for taking the time to read our rebuttal carefully. We appreciate your positive assessment and your constructive feedback throughout the review process.

---

### Official Review · Reviewer_GVEQ · 2026-03-13

**Soundness:** 3
**Presentation:** 2
**Significance:** 2
**Originality:** 3
**Overall Recommendation:** 5
**Confidence:** 4

**Summary:**

The authors present a novel algorithm for semi-supervised learning.  Their method, Two-Teachers
Newton-guided Learning (TTN), leverages a pair of complementary teachers alongside a curvature-aware optimization process.  The author's method seeks to more appropriately utilize pseudo-labels by combining the two teachers' predictions using their confidences as weights.  In this way, a teacher's uncertainty about a prediction is considered.  The authors further propose leveraging to geometry of the loss landscape by weighting the gradient updates using an approximation of the loss landscape's local curvature.  The authors argue that this optimization method promotes smooth and stable convergence.

**Compliance With Llm Reviewing Policy:**

Affirmed.

**Final Justification:**

My final recommendation is for acceptance because I believe the author's work provides an improvement for the optimization of semi-supervised learning methods which can be leveraged to improve a broad number of current methods.  Their contribution is orthogonal to many popular directions and significant as it appears to greatly reduce the required training time for existing methods in a principled way.  The strength of the originality of the approach contributed most significantly to my evaluation.   The strengths in terms of significance also seem notable.  My primary concerns were about the presentation and clarity, but the authors have addressed those concerns in their rebuttal. The author's rebuttal addressed my concerns and improved my impression of their paper.  I have updated my score to reflect this change in impression.

**Key Questions For Authors:**

1. Does the Newton update significantly reduce the time to reach speed for your method?  The authors analysis in Section 4 seems to indicate that this would be the case.  This could be a valuable aspect of the algorithm if it were empirically validated.

**Limitations:**

Yes

**Strengths And Weaknesses:**

Strengths
- Soundness
	- The authors conduct comprehensive ablation studies.  The authors also present compelling evidence that their algorithm is addressing the observed negative aspects of the optimization of prior work.
	- The authors conduct a range of experiments on popular semi-supervised learning datasets.  Their method demonstrates state-of-the-art performance on the most challenging scenarios when access to labels is most limited.
- Originality
	- The authors identify a creative research gap in the semi-supervised learning literature as well as an inventive solution which is well-justified with theoretical insights and empirical evidence.
	- The authors present an interesting idea which aims to advance an important sub-field of machine learning.  While the performance of their method is not superior on all benchmarks to all previous methods they illuminate a direction which is complementary to many directions in prior work and could facilitate future advancements.

Weaknesses

- Significance
	- While it is clear that in all cases TNN is competitive, TTN does not provide significant improvement in several benchmark dataset scenarios.  It is unclear from the experimental results that TTN represents a significant step forward for SSL algorithms.
- Presentation
	- Text elements on Figures 1 and 3 are hard to read.  Please try to keep figure text size on the order of the font size of the manuscript.

---

> ### Author Rebuttal · Authors · 2026-03-30
>
> ***Significance***
> We sincerely thank the reviewer for this thoughtful comment. We agree that the numerical gains of TTN are not uniformly large across every benchmark setting. However, we would like to clarify that our contribution is not simply to obtain a large margin on all datasets by using a stronger teacher pair. Rather, the key novelty of TTN is a new supervision paradigm: moving from conventional zero-order pseudo-label supervision to second-order supervision with curvature-aware optimization.
>
> This distinction is important because progress on modern SSL benchmarks is often incremental, especially under very limited-label settings. In such cases, algorithmic advances should be judged not only by average accuracy gains, but also by stability, robustness, and the ability to improve consistently across different teacher combinations. As shown in Fig. 5, TTN is not tied to a specific pair of teachers such as MAE and DINOv3, and multiple teacher combinations can bring improvements. This suggests that the benefit comes from the proposed training mechanism rather than from a particular model choice.
>
> Therefore, we believe TTN represents a meaningful step forward not only in terms of competitive performance, but also in introducing a more stable and general optimization framework for SSL.
>
>
> ***Presentation***
> Thank you for pointing this out. We agree that the text in Figures 1 and 3 is currently too small for comfortable reading. We will revise both figures and enlarge the text elements so that their font size is consistent with the manuscript and easier to read.
>
> **Q1. *Empirical Validation of Faster Convergence with Newton Updates.***
>
> Additional clarification on the timing analysis. All timing results are reported under a controlled and identical setup for each dataset. Specifically, experiments were conducted on a single NVIDIA GeForce RTX 5090 GPU, using the same implementation, data preprocessing, augmentation pipeline, optimizer setting, and mini-batch configuration across all compared variants. In each iteration, we use a batch size of 32 labeled samples and 32 unlabeled samples. The reported Time / Iter. denotes the average wall-clock training time per iteration under this fixed setup, and Time to X% is computed accordingly from the number of steps required to reach the target accuracy. These timing results are intended for controlled within-method comparison between TTN (full) and TTN w/o Newton (aligned with the experimental settings in Table 4.), rather than as absolute cross-paper speed benchmarks, since wall-clock time is inherently affected by hardware and implementation details. Under this matched setting, the Newton-guided variant consistently reaches the same target accuracy in fewer steps, while maintaining nearly the same per-iteration cost, because the dominant overhead comes from the dual frozen teacher forward passes rather than the Newton update itself.
>
> The following tables provide an empirical validation of the convergence benefit of Newton updates under the controlled setup described above.
>
> **CIFAR-100 (400 labeled)**
>
> | Method         | Final Acc. (%) | Step to 75% | Time to 75% |  Step to 80% |           Time to 80% | Time / Iter. |
> | -------------- | -------------: | ----------: | ----------: | -----------: | --------------------: | -----------: |
> | TTN (full)     |           81.4 |       2.8k | 2.9 ks |        6.0k  |           6.3 ks |      0.992 s |
> | TTN w/o Newton |         80     |       4.0k | 4.1 ks |       11.0k  | 10.5ks           |      0.973 s |
>
>
> **CIFAR-10 (4k labeled)**
>
> | Method             | Final Acc. (%) | Step to 85% | Time to 85% | Step to 89% |  Time to 89% | Step to 93% | Time to 93% | Time / Iter. |
> | ------------------ | -------------: | ----------: | ----------: | ----------: | -----------: | ----------: | ----------: | -----------: |
> | TTN (full)     |       97.5 |   2.2k | 2178 s |   4.0k |  3960 s |   6.5k | 6435 s |  0.987 s |
> | TTN w/o Newton |       89.8 |   4.8k | 4656 s |  12.5k | 12125 s |     N/R |     N/R |  0.969 s |

---

> > ### Author Rebuttal · Reviewer_GVEQ · 2026-04-02
> >
> > I thank the authors for the additional analysis on the time to convergence for their method.  I think this additional information strengthens the value proposition of TTN.  Based on the author's comments I better understand the nature of their contribution and how it fits within the SSL literature.  I agree that there are other important aspects to consider than simply end-task accuracy when evaluating the contributions of a SSL algorithm.  The authors have adequately addressed my concerns.

---

> > > ### Author Response · Authors · 2026-04-02
> > >
> > > Thank you very much for your thoughtful follow-up and positive reassessment. Since you indicate that your concerns have been fully resolved, we would sincerely appreciate it if you could consider updating your score accordingly. Thank you again for your careful reading and constructive feedback.

---

### Official Review · Reviewer_pQce · 2026-03-14

**Soundness:** 3
**Presentation:** 2
**Significance:** 2
**Originality:** 2
**Overall Recommendation:** 3
**Confidence:** 4

**Summary:**

This paper presents TTN (Two-Teachers Newton-guided Learning), a dual-teacher framework
that integrates complementary supervision from MAE and DINOv3 and optimizes the student
through a Newton step update. The experiments show the improvements over single-teacher methods.

**Compliance With Llm Reviewing Policy:**

Affirmed.

**Key Questions For Authors:**

1. The TTN method uses two teachers to construct the dual teacher framework. But the supervision is from two self-supervised models, as MAE and DINOv3. What about the other combination of other models (Figure 5 is somewhat confusing)? Especially, DINOv3 is trained with a much larger dataset. Does the data size in teachers benefit the performance improvement? The effects of the proposed network need to be decoupled from the dependent teacher models.
2. This work uses Newton updates, which is interesting. What is the advantage of Newton updates? Would the Newton updates accelerate the learning, similar to the Newton descent versus gradient descent?
3. Table 4 presents the ablation study. For single-teacher learning, how to compute the two teacher consistency?

**Limitations:**

Yes

**Strengths And Weaknesses:**

Strengths
The idea of introducing two teachers with Newton updates is interesting.
The paper is well written and easy following

Weakness
The improvement is somewhat promising but insignificant.
The design is restricted with two specific teachers, as MAE and Dinov3.

---

> ### Author Rebuttal · Authors · 2026-03-30
>
> We sincerely appreciate the reviewer’s expertise and thoughtful comments. We understand the concern that our current implementation uses MAE and DINOv3 as the two teachers. However, our method is not restricted to these two specific models. The proposed framework is general and can be built on different vision foundation models (VFMs). As shown in Fig. 5, multiple teacher combinations lead to consistent improvements, which suggests that the framework does not depend on one particular pair of teachers, but can benefit from different combinations of representations.
>
> The main contribution of our work is not simply to replace one teacher with another stronger VFM teacher. Our key contribution is a shift in the supervision paradigm, from zero-order supervision, based only on pseudo-label outputs, to second-order supervision that also incorporates curvature-aware optimization. We believe this conceptual change is more important than a small numerical gain alone.
>
> In modern SSL benchmarks, performance improvements are often incremental. Therefore, under extremely limited-label settings, training stability and robustness are more meaningful indicators of algorithmic progress. In this sense, our method contributes not only competitive accuracy, but also a more stable and reliable optimization process.
>
>
> **Q1. *Performance depends on the choice of teacher model.***
>
> We agree that teacher pretraining scale may influence performance to some extent. However, the gains of TTN cannot be attributed to data scale alone. As shown in Fig. 5, not all teacher pairs with strong pretrained models lead to comparable improvements, and some combinations even show weak or negative transfer. If teacher scale were the main factor, the improvements should be more uniform across pairs. For example, a model trained by a teacher in combination with DINOv3 should have equally good performance. Instead, the results indicate that TTN mainly benefits from complementary representations across teachers rather than stronger pretraining alone.
>
> **Q2. *The advantage of Newton updates.***
>
> The practical advantage of the Newton-guided update in TTN is a lightweight diagonal curvature preconditioning step derived from fused teacher geometry. This improves optimization stability by suppressing updates along sharp directions. Aditionally, under the assumptions of Theorem 4.6, this also improves local convergence conditioning relative to an unpreconditioned first-order update. We further clarify this from the following three aspects :
>
> (1) Improved stability.
> Unlike standard gradient descent, the update is preconditioned by the fused inverse Hessian, which rescales gradients according to local curvature. This suppresses updates along sharp directions and avoids oscillatory behavior. Empirically, Fig. 3 shows that TTN exhibits significantly smoother loss trajectories and stable pseudo-label utilization, in contrast to the strong oscillations observed in conventional SSL methods.
>
> (2) Faster and more efficient convergence.
> Theoretically, Theorem 4.6 shows that the Newton-style update is governed by the condition number of the preconditioned curvature matrix $P^{-1}H$, rather than that of the raw Hessian $H$, where  $P$ is the diagonal preconditioner derived from the fused teacher geometry. Since  $P$ provides a better local curvature scaling,  $P^{-1}H$ is better conditioned, which leads to faster local convergence than an unpreconditioned first-order update. Empirically, we also observe more stable and faster loss reduction during training.
>
> (3) Strong empirical evidence from ablation studies.
> The importance of the Newton update is clearly validated in Table 4: removing the Newton step causes a substantial performance drop from 97.5% to 89.8% (-7.7%), which is the largest degradation among all components. This demonstrates that second-order preconditioning is critical for both optimization stability and final performance.
>
>
> **Q3. *Explanation of Table 4.***
>
>
> We thank the reviewer for this question and would like to clarify this point.
>
> In Table 4, “Consistency” refers to the augmentation consistency term $\mathcal{L}_{\mathrm{cons}}$​ in Eq. (10), not a consistency term between two teachers. For the single-teacher variant, the same loss​ is still used, while the dual-teacher fusion branch is removed. Therefore, no teacher-teacher consistency is computed in the single-teacher case.
>
> Importantly, this design ensures that the performance gap between single- and dual-teacher settings (Table 4) reflects the contribution of complementary supervision and curvature fusion, rather than any artificial consistency constructed from a single teacher.
>
> We will revise the table description to clarify this point and avoid confusion.
>
> We hope that our responses help clarify your questions and showcase the potential of our work. If so, we kindly invite you to consider raising your score .

---

### Decision · Program_Chairs · 2026-04-30

**Decision:**

Accept (regular)

**Comment:**

After the author-reviewer discussion period, two of the reviewers fully supported the paper for acceptance. Although the third reviewer was not completely satisfied by the responses, their overall impressions remained positive. Furthermore, the AC finds that the questions in reviewer pQce's initial review were well-addressed. Thus, the AC recommends the paper for acceptance.